# MiWEndo: Evaluation of a Microwave Colonoscopy Algorithm for Early Colorectal Cancer Detection in Ex Vivo Human Colon Models

**DOI:** 10.3390/s22134902

**Published:** 2022-06-29

**Authors:** Marta Guardiola, Walid Dghoughi, Roberto Sont, Alejandra Garrido, Sergi Marcoval, Luz María Neira, Ignasi Belda, Glòria Fernández-Esparrach

**Affiliations:** 1MiWEndo Solutions S.L., 08014 Barcelona, Spain; wdghoughi@miwendo.com (W.D.); rsont@miwendo.com (R.S.); agarrido@miwendo.com (A.G.); smarcoval@miwendo.com (S.M.); lmneira@miwendo.com (L.M.N.); ibelda@miwendo.com (I.B.); gfernan@miwendo.com (G.F.-E.); 2Endoscopy Unit, Gastroenterology Department, Hospital Clínic, University of Barcelona, 08036 Barcelona, Spain

**Keywords:** endoscopes, medical diagnostic imaging, microwave antenna arrays, microwave imaging, colorectal cancer

## Abstract

This study assesses the efficacy of detecting colorectal cancer precursors or polyps in an ex vivo human colon model with a microwave colonoscopy algorithm. Nowadays, 22% of polyps go undetected with conventional colonoscopy, and the risk of cancer after a negative colonoscopy can be up to 7.9%. We developed a microwave colonoscopy device that consists of a cylindrical ring-shaped switchable microwave antenna array that can be attached to the tip of a conventional colonoscope as an accessory. The accessory is connected to an external unit that allows successive measurements of the colon and processes the measurements with a microwave imaging algorithm. An acoustic signal is generated when a polyp is detected. Fifteen ex vivo freshly excised human colons with cancer (*n* = 12) or polyps (*n* = 3) were examined with the microwave-assisted colonoscopy system simulating a real colonoscopy exploration. After the experiment, the dielectric properties of the specimens were measured with a coaxial probe and the samples underwent a pathology analysis. The results show that all the neoplasms were detected with a sensitivity of 100% and specificity of 87.4%.

## 1. Introduction

Colorectal cancer (CRC) is a major health and economic burden in the context of an increasingly aging population. Globally, 1.93 million new cases of CRCs are diagnosed each year, and 935,000 people died of CRCs in 2020 [1], making it the second most common cause of cancer death in both men and women. CRC is a malignant disease that affects the colon and rectum. Ninety percent of CRCs start as a polyp, i.e., an abnormal colon mucosa growth [2]. Although some polyps are harmless, the adenoma type is premalignant and slowly becomes cancerous. Since polyps are usually asymptomatic or present mild symptoms, CRC is commonly found in advanced stages [3]. The five-year survival rate is approximately 65%, but it strongly depends on the development stage at diagnosis, dropping to 14% if cancer has spread to distant parts of the body. Fortunately, early diagnosis can dramatically improve prognosis, saving lives and reducing healthcare costs.

To date, colonoscopy is the most effective diagnostic and therapeutic technique for preventing colorectal cancer since it allows the identification and removal of polyps in the entire colon. Several prospective studies demonstrate that colonoscopy with polypectomy reduces the incidence of CRC by 40–90% [4,5]. Nevertheless, colonoscopy is far from perfect: 22% of polyps go undetected [6], and cancer incidence after a negative colonoscopy is still 7.9% [7]. This lack of efficacy is mainly caused by visualization limitations, since the camera has a field of view of less than 180° [8], inhomogeneous illumination, colon angulations and folds, and poor colon cleaning. For this reason, 13.4% of the colon surface area might not be visualized during a colonoscopy [9].

Several endoscopic devices and technologies have been developed to improve the adenoma detection rate (ADR). ADR is the quality indicator of colonoscopy, as each 1% increase in ADR decreases patients’ risk of CRC by 3% [10]. High-definition endoscopes, endoscopes with multiple lenses, and mucosal flattening accessories [11,12] have demonstrated an increase in ADR of 4.5%, 5%, and 16%, respectively, by improving colon visualization [13]. Artificial intelligence for the real-time assessment of endoscopic images [14] has shown an increase in ADR of 14% [15]. However, if the camera does not visualize the adenoma, the algorithm cannot detect it. Chromoendoscopy, endoscopic microscopy (endocytoscopy and endomicroscopy), and hyperspectral techniques [16] are methods that have been developed to magnify and enhance mucosa tissue characteristics linked to malignancy, but have failed to demonstrate significant increases in ADR [13]. All these techniques are restricted to the optical information captured by the camera and require highly trained professionals. Their outcomes are highly dependent on the operator’s experience and human factors like fatigue, stress, and resilience. Therefore, a tool capable of automatizing the detection of polyps is needed.

We recently proposed microwave imaging, a non-ionizing, anatomical, and functional imaging method [17] to detect polyps in the colon. Compared to advanced colonoscopy methods, microwave imaging offers a new contrast mechanism that is independent of the information captured by the optical camera. Microwave imaging retrieves the dielectric properties of a target—the relative permittivity and the conductivity—from the measured electromagnetic fields. The dielectric properties are biomarkers of several health conditions, such as breast cancer [18,19], brain stroke [20], osteoporosis [21], heart infarction [22], or edema. We demonstrated in an ex vivo study with human colon polyps and healthy mucosa that the dielectric properties of colon polyps increase when the malignancy grade increases. At 8 GHz, the contrast between healthy colon mucosa and colorectal cancer is 30% and 90% for the relative permittivity and conductivity, respectively [22]. We developed a prototype to integrate microwave imaging into conventional colonoscopy called MiWEndo. MiWEndo is a ring-shaped switchable antenna array accessory attachable to the distal tip of a standard colonoscope connected to an external processing unit. MiWEndo provides a field of view inside the colon of 360° and generates an acoustic signal when it detects a polyp in the microwave image [23]. A frequency-domain reconstruction scheme based on a modified back-projection method preceded by a calibration was developed to form cross-sectional images of the colon every 4 mm for each colon model. A fixed threshold was used to detect if a polyp was present in each image. A preliminary algorithm was tested in phantoms [24] and ex vivo models [25]. These studies concluded that the calibration and the detection steps had to be improved. In [26] we performed a comparative study between five different calibration methods to define the most suitable one.

In this paper we present an improved microwave colonoscopy algorithm for early colorectal cancer detection. This study aims to assess the overall algorithm performance with 15 freshly excised human colon specimens with neoplastic lesions. This algorithm uses the calibration scheme identified in [26] and an improved automatic detection method. The colon models were placed on a setup designed to emulate a colonoscopy exploration with the MiWEndo microwave-based colonoscopy system. The results were compared with the gold standard pathology analysis in terms of sensitivity and specificity. We also characterized the dielectric properties of the specimens at 7.6 GHz with an open-ended coaxial probe.

## 2. Materials and Methods

### 2.1. Microwave-Based Colonoscopy System

The imaging system is composed of an external processing unit and an acquisition accessory. The external unit contains a vector network analyzer (E5071C ENA VNA, Keysight, Santa Rosa, CA, USA), a microcontroller (Arduino Nano, Arduino, New York, NY, USA), and a laptop. The acquisition accessory consists of a ring-shaped encapsulated switchable antenna array that can be attached to a conventional colonoscope’s tip, as shown in Figure 1. The antenna array comprises two rings of eight antennas, one containing the transmitting and the other the receiving antennas. The antenna elements are cavity-backed slot antennas operating over the 7.6–7.66 GHz range in free space. Figure 2 shows the antenna geometry and S-parameters. The reduced bandwidth is due to the miniaturization of the electrically small antennas, 0.16λ×0.12λ, as described in a previous paper [23]. The frequency was chosen because the range from 5 GHz to 8 GHz is where there is more contrast in dielectric properties between healthy colon tissue and polyps [27]. In this application, using a high frequency is not a problem since polyp detection does not require penetration into the tissue since they are superficial lesions. The output power of the VNA is −5 dBm, due to the attenuation of the cables and the multiplexer, the power arriving to the antennas is between −13 dBm and −15 dBm. As the antennas have an efficiency of 22%, the radiated power is between 0.007 mW to 0.011 mW at 7.5 GHz. The antenna elements are welded onto a polyamide flexible printed circuit board (PCB) containing microstrip feeding lines and two single-pole-eight-throw (SP8T) radiofrequency switches. This assembly is wrapped around a 3D printed piece that adapts the PCB with the colonoscope. Finally, it is encapsulated with a biocompatible 3D printed resin to protect the PCB from moisture and avoid injuring the colon mucosa tissue. The final dimensions of the accessory are 30 mm in length by 20 mm in diameter, exceeding the colonoscope’s diameter by less than 3 mm. The dimensions and shape of the accessory device ensure non-obstruction of the colonoscope’s front tip, maintain the maneuverability of the colonoscope, avoid camera concealment, and prevent injuries to the patient. Two slim 500 mm in length and 1.13 mm in diameter coaxial cables transmit the microwave signals, and eight wires transmit the switches’ control signals. Miniature connectors are used for a non-bulky final assembly.

### 2.2. Setup for Ex Vivo Human Colon Measurements

The colon is the last part of the gastrointestinal tract. It has a segmented appearance due to a series of folds, and it is about 1500 mm long and 40–90 mm in diameter. Polyps are slow-growing overgrowths of the colonic mucosa protruding into the lumen.

This study was performed at the Pathology Unit of Hospital Clínic de Barcelona and was conducted according to the guidelines of the Declaration of Helsinki and approved by the Institutional Review Board of Hospital Clínic de Barcelona (protocol code HCB/2017/0519 and date of approval 29 June 2017). Informed consent was obtained from all subjects involved in the study. A total of 15 patients undergoing surgical colectomies were enrolled in the study. The protocol, shown in Figure 3, was defined in close collaboration with the medical staff to reduce the manipulation of the samples and time between excision and measurement. The colon samples were obtained from surgical colectomies performed in an operating room with the shortest possible cold ischemia time and without any conservation treatment (i.e., without formaldehyde) to preserve their physical and dielectric properties. Samples from patients that underwent previous radiation therapy or chemotherapy were excluded. Next, a pathologist opened the colon sample containing a neoplasm longitudinally, placed it in a hermetically sealed plastic container, and transported it to the measurement room. Immediately after sample arrival, we performed the exploration with MiWEndo’s accessory. To reproduce a realistic colonoscopy exploration with ex vivo colon fragments, we created a colon fixation setup and a positioning setup to move MiWEndo’s accessory along the ex vivo colon lumen shown in Figure 4a. The positioning setup was composed of a T-shaped metallic structure fixed on a plastic base. The metallic structure holds a plastic bar that simulates a colonoscope tube. MiWEndo’s accessory was attached at the tip of the plastic bar. The colon sample was fixed around a tube of expanded polystyrene and externally wrapped with a soaker pad. Expanded polystyrene behaves like air for microwaves and therefore has a minimal impact on the radiated electromagnetic fields. It represents a realistic situation since, during a colonoscopy, the colon is expanded using carbon dioxide insufflation to increase colon visualization. The wrapped sample was placed on a platform with L-shaped plastic pieces to fix the sample during the procedure, and the platform was placed on a lifting platform to control the vertical position. Finally, the lifting platform was placed next to the positioning setup. The plastic bar with the MiWEndo’s accessory attached at its tip was manually pushed inside the tube of expanded polystyrene that models the colon lumen in steps of 4 mm. A measurement was made before each step until reaching the end of the sample. Table 1 shows the characteristics of the human colon samples measured. Samples 11, 12, and 13 were wrapped around a curved polystyrene foam to model a colon angulation, simulating a more challenging situation. Moreover, Sample 12 presented a suboptimal cleaning of the colon with some debris along the mucosa.

After exploring the colon sample with MiWEndo’s accessory, the sample was removed from the measurement setup. With the help of the pathologist the position position of the lesion with respect to the beginning of the trajectory was registered. These data constitute the ground truth necessary to evaluate the algorithm’s performance. Next, the dielectric properties of the colon mucosa and the neoplasm were measured using the open-ended coaxial probe method using the N1501A Dielectric Slim Form Probe Kit, Keysight, Santa Rosa, California, US with the N1500A Materials Measurement Software Suite version 16.0.16092801, Keysight, Santa Rosa, CA, USA connected to a vector network analyzer Keysight E5071C. We performed three measurements on the neoplasm and three measurements on three equidistant locations on the healthy mucosa (start, center, and end). The different tissues were identified by a pathologist prior to the measurement with the coaxial probe. We followed the recommended procedure for obtaining high quality measurements with the open ended coaxial probe [27,28]. The setup is depicted in Figure 4b. The entire protocol was performed between 30 min and one hour after colon resection. Finally, the specimens were sent to the Pathology Department for the histological analysis of the neoplasms.

### 2.3. Polyp Detection Algorithm for Microwave-Assisted Colonoscopy

The MiWEndo’s accessory was moved inside the colon model in a straight trajectory throughout the center of the lumen to simulate a colonoscopy exploration for polyp detection. A microwave image of a cross-section of the colon (XY plane) was obtained for each step in the trajectory, which was performed in 4 mm increments along the z-axis. Figure 5 presents the geometry of the imaging setup. Each step in the trajectory and its microwave image is referred to as a frame, being n the index of the current frame.

The data for creating the microwave image at each frame is obtained by subsequently illuminating the colon tissue with a microwave signal radiating from a transmitting antenna and measuring the total field at the three receiving antennas that are closest to the transmitting one (the adjacent antenna and the two diagonal antennas). The measured total field results from the interaction of the incident microwaves and the colon tissues and therefore contains information about the spatial distribution of the dielectric properties of the colon. The process is repeated for each of the 8 transmitting antennas to scan the entire perimeter of the colon. In total, 24 combinations of transmission S-parameters are measured and used to produce the microwave image the frame’s cross-sectional slice of the colon.

The microwave image reconstructions were obtained at a single frequency of 7.6 GHz corresponding to the center frequency of the antennas. Since the antennas are narrowband, using more frequencies within their bandwidth does not provide additional independent information for reconstruction.

A three-step algorithm is used to obtain the output, i.e., the acoustic signal when a polyp is detected, for each measured frame n. 

#### 2.3.1. Calibration

The aim of the calibration step is to clean the total field of unwanted effects including: the unknown distance to the colon walls, the colon folds, the angulations, etc., and to keep the response due to the presence of the polyp. To fulfill this purpose, a calibration strategy, which was given the name of automatic temporal (AT) subtraction calibration, was implemented [26]. The AT subtraction calibration relies on the hypothesis that the impact on the scattered field that is caused by a polyp is much more important than the one produced by any other of the previously mentioned effects. Hence, for each frame, this strategy identifies, from the pool of previous frames, the frame with the most similar scattered field and subtracts it. By doing so, the only effect that would remain after the calibration is the polyp response, while the rest of undesirable effects would be minimized since they have a similar effect on the subtracted frames. The similarity between frames is computed using the modified Hausdorff distance, whose potential as S-parameters similarity measurer has already been proven [29]. Therefore, the frame with the most similar S-parameters to the current one is computed as follows: (1)$$S^{\prime }:\max(∥S(n)-S^{\prime }∥))=\min(\max(∥S(n)-{S^{\prime }}_{i}∥))\;for\;i=1,\;\ldots ,\;n-1$$
where S(n) corresponds to the frame that is being calibrated, S’ to the most similar frame to S(n), and S’_i_ to the previously acquired frames.

#### 2.3.2. Focuser

Once the data is calibrated, a modified monofocusing algorithm [30] is applied to obtain an image of the dielectric contrast profile $I_{z_{n}}\left(\vec{r}\right)$:(2)$$I_{z_{n}}\left(\vec{r}\right)=\left|\overset{{\left(i+1\right)}_{N_{a}}}{\underset{j={\left(i-1\right)}_{N_{a}}}{\sum }}\overset{N_{a}-1}{\underset{i=0}{\sum }}E_{s}^{2}\left({\vec{r}}_{T_{i}},\;{\vec{r}}_{R_{j}},\;z_{n}\right)J_{1}^{2}\left(k\left|{\vec{r}}_{R_{j}}-\vec{r}\right|\right)e^{j2\left(k\left|{\vec{r}}_{R_{j}}-\vec{r}\right|+\varphi \right)}\right|$$
where k=2πcf is the wavenumber, φ is the angle between the transmitting and the receiving antennas, and $E_{s}$ is the scattered electric field. This focuser method provides the image of the dielectric contrast profile. The constant c is the speed of light since the colon is insufflated with carbon dioxide during the procedure.

#### 2.3.3. Detector

The detection method implemented is based on a forecasting exponential smoothing method for the detection of outliers [31,32]. We have observed that high amplitudes in the microwave images usually correspond to the presence of a polyp, but the high amplitude values are not comparable between different trajectories (different colon specimens). Therefore, we can detect the presence of a polyp by finding outlying values of the maximum amplitude of the reconstructed microwave image compared to previous frames in the same trajectory. 

The first step in the detection algorithm is to compute an estimate of the maximum amplitude of the image for the next frame, $s\left(n+1\right)$, based on the previous estimation of the current frame and its actual maximum amplitude, $y\left(n\right)$, as follows:(3)$$s\left(n+1\right)=\alpha \times y\left(n\right)+\left(1-\alpha \right)\times s\left(n\right)$$
where α is a constant between [0, 1]. The next step is to compute the confidence band b, in which the next maximum amplitude of the image will likely lie.
(4)$$b\left(n+1\right)=s\left(n+1\right)\;\pm \;\left[\alpha \times \left[y\left(n\right)-s\left(n\right)\right]+\left(1-\alpha \right)\times b\left(n\right)\right]\times \delta$$

Here δ is a constant higher than 1. When the maximum reconstructed amplitude of the analyzed frame lies above its confidence band, the algorithm concludes that a polyp is detected and activates the acoustic signal.

## 3. Results

### 3.1. Sample Characterization Results

Using the data provided in the pathology report, we identified and labeled all the measured frames: healthy mucosa, adenomas with high grade dysplasia (HGD), and adenocarcinomas. Note that high grade dysplasia refers to precancerous changes in the cells. Table 1 summarizes the main characteristics of the 15 colon examinations, and Figure 6 shows the relative permittivity and conductivity of the healthy colon mucosa and neoplasm of each patient. If we look at the dielectric properties of each patient individually, we see that the dielectric properties of the neoplasm are always higher than those of the healthy mucosa. However, there is considerable variability between patients. This makes it impossible to establish a threshold that allows the differentiation between the healthy mucosa and the neoplasm for all patients. It is worth mentioning that accurately measuring the dielectric properties of healthy mucosa is very difficult since it is a very fine tissue, between 0.5 and 2 mm. Moreover, the part opposite the lumen was covered with very variable amounts of fat. Since we could not manipulate the samples so as not to alter the subsequent pathological analysis, the uniformity of the samples cannot be guaranteed. So, part of the variability between patients could be attributed to this issue.

### 3.2. Polyp Detection Results

Following data acquisition and processing, the results of the microwave-based colonoscopy were obtained. The performance of the method to detect polyps in a trajectory is evaluated using the sensitivity and specificity. The sensitivity, also called the true positive (TP) rate, measures the percentage of cases having a polyp that are correctly diagnosed as having the lesion. A false negative (FN) occurs when a negative result is reported to a trajectory that does have a polyp. The specificity, also called the true negative (TN) rate, measures the percentage of healthy cases that are correctly identified as not having any polyp. A false positive (FP) is reported when the test wrongly indicates that a polyp is present. The values of sensitivity and specificity are related to TP, FP, TN, and FN values as follows:(5)$$\mathrm{Sensitivity}\;\left(\%\right)=\frac{\mathrm{TP}}{\mathrm{TP}+\mathrm{FN}}\;\;,\;\;\;\;\;\;\;\;\mathrm{Specificity}\;\left(\%\right)=\frac{\mathrm{TN}}{\mathrm{TN}+\mathrm{FP}}\;\;$$

Table 1 shows that in all the 15 cases the neoplasm has been detected with microwave-based colonoscopy. The overall sensitivity is 100% and the specificity is 87.43%.

As an example, Figure 7a shows the evolution of the normalized maximum amplitude of the reconstructed image registered in each step of the trajectory, where it is possible to appreciate the increase of amplitude due to the presence of the lesion. The reconstructed image has been obtained using Equation (2). Figure 7c presents the normalized maximum amplitude of the microwave image in front of each pair of antennas, and, here, it is moreover possible to observe that the impact of the lesion is much more significant on the antennas 0–7, 0, 1 (Transmitter 0 and Receivers 7, 0, and 1), which are the ones that face the neoplasm. Overall, both figures demonstrate that the presence of the lesion produce an increase in the amplitude of the reconstructed images, which is not registered in the healthy tissue.

## 4. Discussion

Missed lesions at colonoscopy are as high as 22% and have important clinical consequences. In the last years, many efforts have been made to improve the performance of endoscopy, but the common limitation of these methods is that they cannot detect what is not displayed by the camera. 

In this study we have shown that the improved version of the algorithm is able to detect neoplastic lesions in human colon models. To do so, the MiWEndo microwave colonoscopy accessory device attached to a colonoscopy simulating setup was used. The system can detect neoplastic lesions on the full perimeter of the colon based on the changes in their dielectric properties and may be used to complement the endoscopic image. The system has been designed to be compatible with colonoscopy, ensure a full coverage and produce minimal changes to the current clinical practice. The final dimensions of the acquisition accessory device are 30 mm in length by 20 mm in diameter, with a total thickness of 3 mm. The dimensions and shape of the device ensure non-obstruction of the front tip of the colonoscope, avoiding injuring the patient or hindering the maneuverability of the colonoscope. The antennas work at 7.6 GHz over a narrow bandwidth and with an acceptable coupling. 

A frequency domain imaging reconstruction algorithm based on Fourier backprojection preceded with an automatic temporal subtraction calibration that has been developed to obtain cross sectional images of the colon. An automatic detection step based on exponential smoothing was applied to obtain an output easy to interpret by the physicians, i.e., and acoustic alarm, when a polyp is detected. This kind of output was inspired by the acoustic parking system for automobiles. The system provides an audible warning with a series of beeps as the car, or the endoscope, moves nearer a stationary object, or a polyp. The algorithm is fast and able to work in real time, delivering around 10 frames/s. In terms of performance, the algorithm has reached a sensitivity of 100% and a specificity of 87.43%. No significant degradation of performance has been observed when the trajectory is not straight (trajectories 11, 13, and 14). This may indicate that the algorithm, and especially the calibration, can deal with the angulations of the colon. Debris in the colon (trajectory 12) did not degraded performance either. In the absence of further validation and to confirm this in future tests, it could be examined as to whether microwave-assisted colonoscopy could reduce the need for colon preparation. The impact of poor preparation is important since it forces to repeat the colonoscopy. In addition, the preparation required for the patient to clean the colon is the most uncomfortable part of the colonoscopy, according to patients. It is worth mentioning that by changing the value of the constants α and δ used in the exponential smoothing method it is possible to tune the sensitivity and specificity values. We decided to force an optimal sensitivity because our device is aimed to prevent polyps from being missed once combined with colonoscopy. A suboptimal sensitivity can be accepted because our device is used in conjunction with the colonoscope. In cases where the acoustic signal is heard, the finding will always be confirmed with the video from the colonoscope.

Regarding the dielectric and pathologic characterization of the lesions, at 7.6 GHz healthy colon mucosa and neoplastic lesions exhibit inter-patient contrast in dielectric properties that allow its differentiation. However, the variation between patients do not allow to fix a permittivity threshold to differentiate all the healthy and neoplastic tissues. This is not a problem for the MiWEndo’s system since the detection criteria is defined for each patient. 

The main limitation of this study is the low number of adenomas studied. The device was validated in ex vivo human colons that usually are surgically resected because they have already developed cancer. These lesions are commonly big and are not representative of what is commonly found in a regular colonoscopy. The second limitation is that the real-time acquisition system has not been tested. In this study, a measurement was taken after each trajectory step manually. To make the acquisition in real time, a second generation of external unit is being developed. The last limitation is that the device was not attached at the end of a real endoscope and the maneuverability and safety could not be assessed. Safety must be guaranteed on two fronts: energy safety and mechanical safety. In terms of energy, we refer to limit the radiation to avoid any biological effect to the patient’s mucosa, i.e., heating, but also ensure the electrical safety and electromagnetic compatibility. In terms of mechanical safety, the accessory device must be designed with smooth surface, using biocompatible materials in the parts that are in contact with the patient and the materials must be clean, i.e., with low bioburden.

For all these reasons, we are developing increasingly realistic anatomic models with folds and polyps of different sizes and shapes made of materials that mimic the dielectric properties of colon tissues. In the phantoms we also can study different trajectories, varying the distance between the antennas and the colon wall while controlling the ground truth, i.e., the position of the polyp. The results with the simplest phantom can be found in [24]. We are also planning new studies in animal models to check the maneuverability and the inter-compatibility with other endoscopic devices used in real explorations such as forceps or electrocautery tools.

## 5. Conclusions

This paper presents the algorithm validation of a microwave colonoscopy system with 15 ex vivo human colons with neoplastic lesions. The system consists of an acquisition accessory attachable to the distal tip of a standard colonoscope connected to an external processing unit. The algorithm is composed of three steps: calibration, focusing, and automatic detection. A sensitivity of 100% and a specificity of 87.43% was achieved, indicating that the detection algorithm is able to deal with real human colon tissues. The algorithm was able to deal with colon angulations. This confirmed what we observed with phantoms, where a more realistic movement within the colon was reproduced. This demonstrates that the calibration method can deal with the changing distances between the antennas and the colon wall. In one trajectory, the algorithm was able to detect the lesion even in suboptimal cleaning conditions.

We are currently carrying out a measurement campaign with a new phantom extracted from a 3D model of a real patient with angulations and folds. With this model, several trajectories will be measured with the accessory attached to a real colonoscope and the algorithm will be used in real time. This setup will allow the conclusion of the algorithm validation with more challenging conditions before starting the clinical phase.

## Figures and Tables

**Figure 1 sensors-22-04902-f001:**
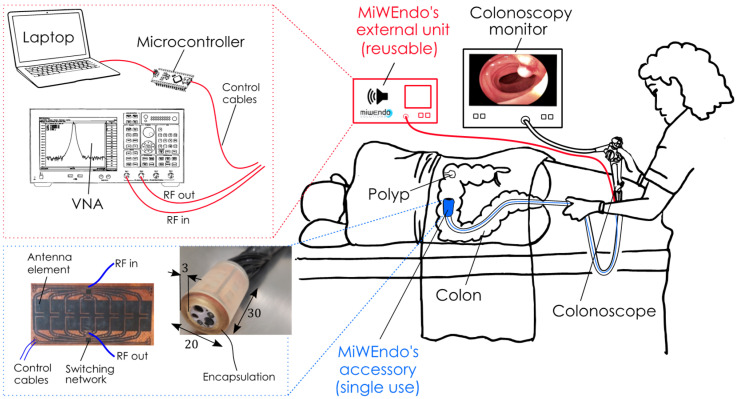
MiWEndo’s microwave endoscopy system is composed by an accessory attachable at the distal tip of a standard colonoscope (**bottom left**) and an external processing unit (**top left**). MiWEndo generates an acoustic signal when a polyp is detected to warn the endoscopist during a colonoscopy exploration (**right**). Encapsulation size is in mm.

**Figure 2 sensors-22-04902-f002:**
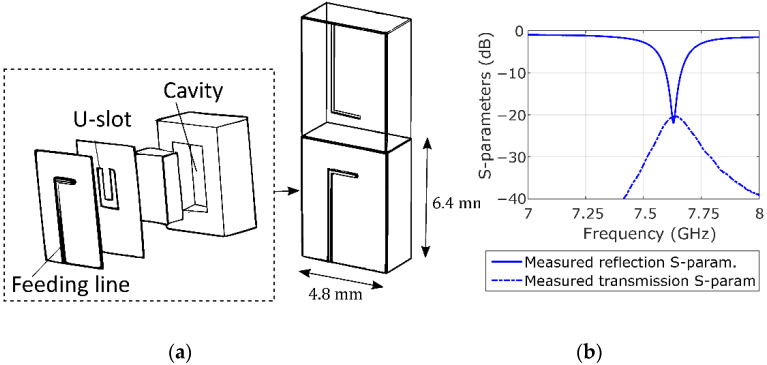
(**a**) Schematic of the cavity-backed slot antenna and a transmitting and receiving antenna pair. (**b**) Measured transmission and reflection S-parameters.

**Figure 3 sensors-22-04902-f003:**
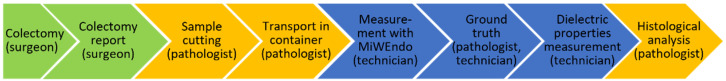
Protocol for the acquisition of electromagnetic measurements within medical routine.

**Figure 4 sensors-22-04902-f004:**
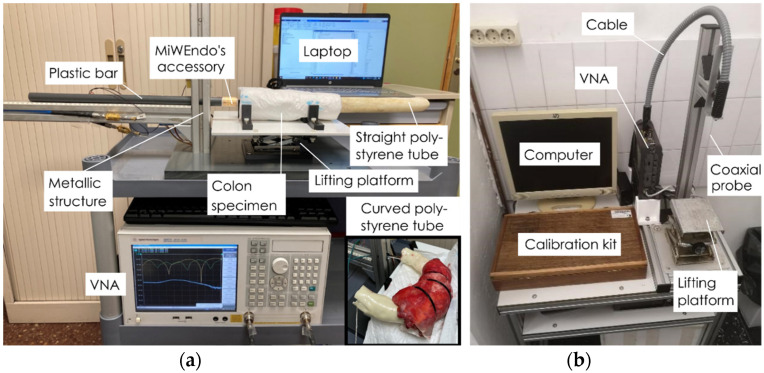
Setup for ex vivo human colon measurements: (**a**) setup for the exploration of ex vivo human colon samples with MiWEndo’s accessory; (**b**) setup for measuring the dielectric properties of ex vivo colon samples with the open-ended coaxial probe method.

**Figure 5 sensors-22-04902-f005:**
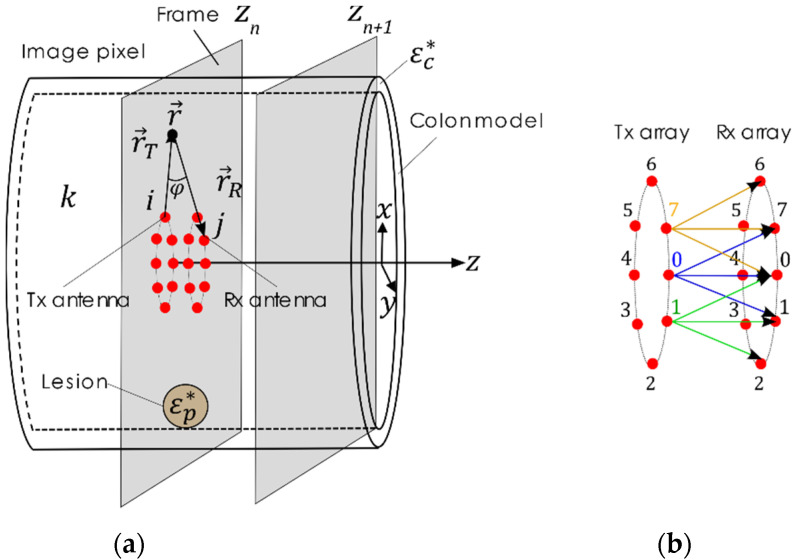
Geometry of the imaging problem: (**a**) schematic of the ex vivo colon model with a lesion of complex permittivity $\varepsilon_{p}{}^{*}$ and the acquisition array moving along the z-direction in the colon lumen. (**b**) Transmitting and receiving array configuration, e.g., for the transmitter i=0, a measurement is taken with the 3 closest receiving antennas, j=7, 0, 1 depicted in blue, for the transmitter i=1 a measurement is taken with antennas j=0, 1, 2 depicted in green, for the transmitter i=7 a measurement is taken with the antennas, j=6, 7, 0 in orange.

**Figure 6 sensors-22-04902-f006:**
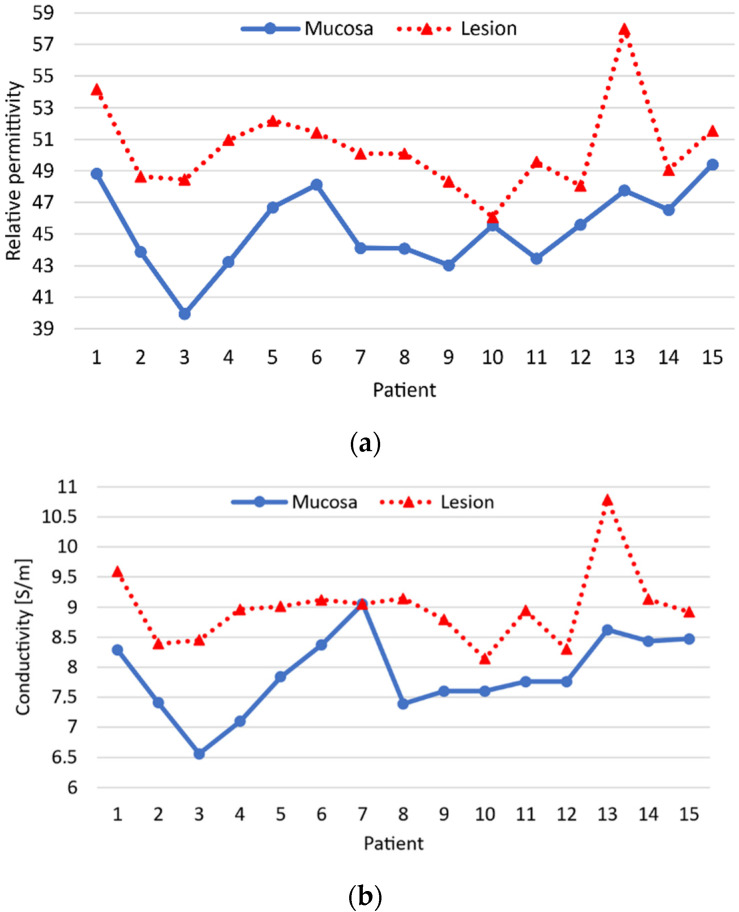
Dielectric properties of the healthy colon mucosa and the neoplasm at 7.6 GHz for each patient was measured with an open-ended coaxial probe. (**a**) Relative permittivity, (**b**) conductivity.

**Figure 7 sensors-22-04902-f007:**
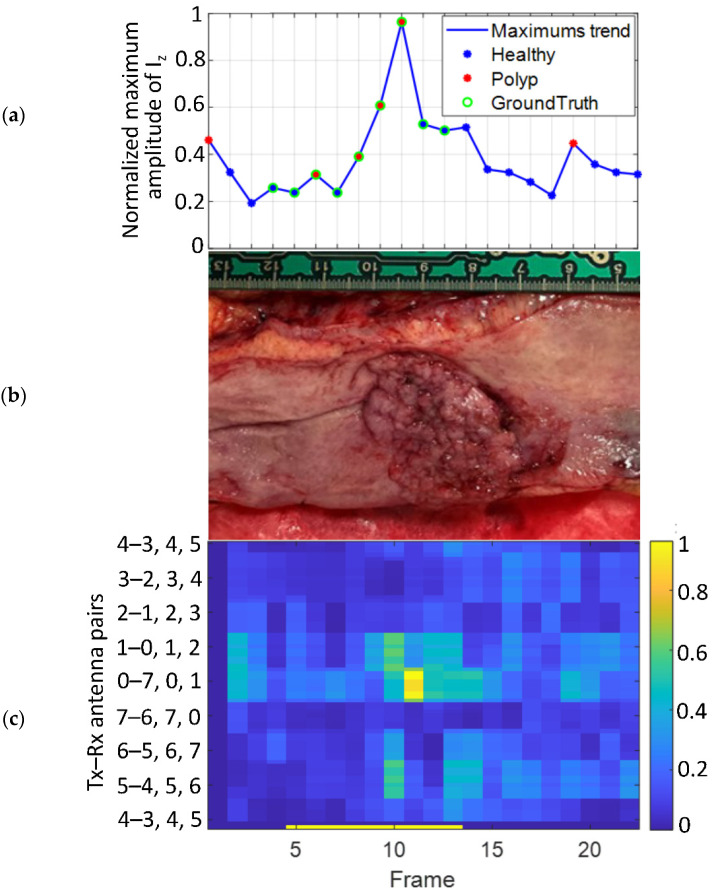
Results of Patient 4. (**a**) The evolution of the normalized maximum amplitude of the reconstructed image for each frame of the trajectory, (**b**) photograph of the colon sample with an adenoma with HGD, (**c**) normalized maximum amplitude in front of all the antenna combinations.

**Table 1 sensors-22-04902-t001:** Characteristics of the trajectories measured in human colon models and performance evaluation of the implemented algorithm in terms of sensitivity and specificity.

Patient	Age	Type of Neoplasm	Neoplasm Size (mm)	Sample Length (mm)	Sensitivity (%)	Specificity (%)
1	86	Adenoma with HGD ^1^	10	200	100	80
2	64	Adenocarcinoma ^2^	50	220	100	94.12
3	46	Adenocarcinoma	36	80	100	100
4	37	Adenoma with HGD	32	155	100	87.50
5	83	Adenocarcinoma	48	190	100	90.91
6	60	Adenocarcinoma	37	190	100	77.78
7	57	Adenocarcinoma	65	330	100	88.37
8	68	Adenocarcinoma	15	320	100	85.71
9	86	Adenoma with HGD	23	270	100	86.54
10	85	Adenocarcinoma	34	285	100	87.23
11	45	Adenocarcinoma ^3^	32	260	100	83.33
12	75	Adenocarcinoma ^4^	35	180	100	86.36
13	91	Adenocarcinoma ^3^	40	160	100	84.21
14	62	Adenocarcinoma ^3^	37	228	100	92.86
15	81	Adenocarcinoma	63	97	100	100

^1^ HGD means high grade dysplasia and refers to precancerous changes in the cells. ^2^ Adenocarcinoma is a cancer that begins in glandular cells. ^3^ Curved trajectories emulating a colon fold. ^4^ Trajectory with a suboptimal colon cleaning.

## Data Availability

All data generated or analyzed during this study are included in this article. Further enquiries can be directed to the corresponding author.

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
