# Peer review of "MiWEndo: Evaluation of a Microwave Colonoscopy Algorithm for Early Colorectal Cancer Detection in Ex Vivo Human Colon Models"

_sensors, 2022, doi:10.3390/s22134902_

Round 1
Reviewer 1 Report
This paper aims to present a novel diagnostic MW-based tool to detect polyps during colonoscopy. Although the authors provide extensive background of the clinical need and current state-of-art, they may improve the description of the experimental setup, hardware and software of the device and presentation of results.
In particular, the paper may be significantly improved if the following comments are addressed.
1) Please, describe how the moisture of the resected colon samples was preserved without biasing the dielectric properties.
2) Add a schematic of the antenna geometry and the array configuration explaining how the data matrix is filled (i.e. 24 combinations of transmission S-parameters).
3) Explain the design choice for the operating frequency being 7.6 GHz. Why narrowband? Is this choice driven by hardware constraints? Please, expand this section.
4) Section 2.3.1 is not clear. It should be completed with refences for Automatic Temporal (AT) subtraction calibration and its formal expression.
5) Figures 5a and 5c should be described better. Please, add the formal expression of the "Maximum reconstructed amplitude" with units. Please, add the unit in fig 5c.
Author Response
Response to Reviewer 1 Comments
Point 1: Please, describe how the moisture of the resected colon samples was preserved without biasing the dielectric properties.
Response 1: We apologize for the lack of detail in the description of sample’s handling during measurements. We updated the manuscript by adding more details on the measurement protocol and a flowchart (Figure 3) with the steps for the ex-vivo colon measurements. We added the following text to clarify the reviewer’s point: “The protocol, shown in Figure 3, was defined in close collaboration with the medical staff to reduce the manipulation of the sample and the time between excision and measurement. The colon samples were obtained from surgical colectomies performed in an operation room with the shortest possible cold ischemia time and without any conservation treatment (i.e., without formaldehyde) to preserve their physical and di-electric properties.” … “Next, the colon sample containing a neoplasm was opened longitudinally by a pathologist, hermetically sealed in a plastic container, and transported to the measurement room. Immediately after sample arrival, we performed the exploration with MiWEndo’s accessory” … “The entire protocol was performed between 30 minutes and one hour after colon resection”.
Point 2: Add a schematic of the antenna geometry and the array configuration explaining how the data matrix is filled (i.e. 24 combinations of transmission S-parameters)
Response 2: A schematic of the antenna geometry has been added in Figure 2 together with its transmission and reflection S-parameters. The array configuration indicating how the data matrix is filled has been added in Figure 5(b).
Point 3: Explain the design choice for the operating frequency being 7.6 GHz. Why narrowband? Is this choice driven by hardware constraints? Please, expand this section.
Response 3: We thank the reviewer for pointing this out. The reduced bandwidth is due to the miniaturization of the antennas that are electrically small. The frequency was chosen because the range from 5 GHz to 8 GHz is where there is more contrast in dielectric properties between healthy colon tissue and polyps. This reasoning has been included in the manuscript in section 2.1.
Point 4: Section 2.3.1 is not clear. It should be completed with refences for Automatic Temporal (AT) subtraction calibration and its formal expression.
Response 4: We agree with reviewer’s comment. We added a reference to a previous paper of ours that presents a benchmark between different calibration methods including AT. We also included the formal expression of AT as suggested by the reviewer.
Point 5: Figures 5a and 5c should be described better. Please, add the formal expression of the "Maximum reconstructed amplitude" with units. Please, add the unit in fig 5c.
Response 5: Following the reviewer's recommendation, we have cited the equation with which the result was obtained, and we have normalized the results presented in figure 5. Since the result has been normalized, the magnitude is dimensionless. Showing the normalized result makes more sense since the reconstruction method is not quantitative and therefore the reconstructed quantity lacks correspondence with any physical quantity.
Please note that we have included an English language and style spell check by a native English-speaking colleague. The revisions are marked up using the “Track Changes” function of MS Word.
Reviewer 2 Report
This study assesses the efficacy of a microwave colonoscopy algorithm using a cylindrical ring-shaped switchable antenna array to detect colorectal cancer precursors or polyps in an ex-vivo human colon model. In general, the work of this paper is well done and has high practical value. However, there are still some problems to be solved, as shown below:
1. Section 2.1. The structure and dimensions of the cylindrical ring-shaped switchable antenna array should be given.
2. Section 2.1. Can you provide the performance parameters of the used antenna, such as gain and return loss?
3. Section 2.1. The working frequency of the antenna is relatively high, and its penetration depth may be relatively small. What is the input power of the antenna? Have you measured the attenuation of microwaves when they reach the colon?
4. Section 2.2. Could you provide a flowchart to introduce the steps for Ex-Vivo Human Colon measurements?
5. Section 2.2. The authors used the ex-vivo colon fragments in the experiment. What are the main differences between ex-vivo colon fragments and the vivo ones?
6. Section 3.1. For the data in Table 1 and Figure 4, were them obtained from another place or by measurement?
Author Response
Response to Reviewer 2 Comments
Point 1:. Section 2.1. The structure and dimensions of the cylindrical ring-shaped switchable antenna array should be given.
Response 1: We appreciate the reviewer’s insightful suggestion. We have added the dimensions of the cylindrical ring-shaped switchable antenna array in Figure 1. To show the structure, we have added the Figure 2(a), that includes a schematic of the cavity-backed slot antenna.
Point 2: Section 2.1. Can you provide the performance parameters of the used antenna, such as gain and return loss?
Response 2: We thank the reviewer for pointing this out. We have added the measured reflection and transmission S-parameters of an antenna pair to show the antenna performance.
Point 3: Section 2.1. The working frequency of the antenna is relatively high, and its penetration depth may be relatively small. What is the input power of the antenna? Have you measured the attenuation of microwaves when they reach the colon?
Response 3: The output power of the VNA is -5 dBm, due to the attenuation of the cables and the multiplexer, the power arriving to the antennas is between -13 dBm and -15 dBm. As the antennas have an efficiency of 22%, the radiated power is between 0.007 mW to 0.011 mW at 7.5 GHz. In this application, using a high frequency is not a problem since the detection of polyps does not require penetration into the tissue since they are superficial lesions. This reasoning has been included in the manuscript in section 2.1.
Point 4: Section 2.2. Could you provide a flowchart to introduce the steps for Ex-Vivo Human Colon measurements?
Response 4: We apologize for the lack of detail in the description of sample’s handling during measurements. We updated the manuscript by adding more details on the measurement protocol and a flowchart (Figure 3) with the steps for the ex-vivo colon measurements. We added the following text to clarify the reviewer’s point: “The protocol, shown in Figure 3, was defined in close collaboration with the medical staff to reduce the manipulation of the sample and the time between excision and measurement. The colon samples were obtained from surgical colectomies performed in an operation room with the shortest possible cold ischemia time and without any conservation treatment (i.e., without formaldehyde) to preserve their physical and di-electric properties.” … “Next, the colon sample containing a neoplasm was opened longitudinally by a pathologist, hermetically sealed in a plastic container, and transported to the measurement room. Immediately after sample arrival, we performed the exploration with MiWEndo’s accessory” … “The entire protocol was performed between 30 minutes and one hour after colon resection”.
Point 5: Section 2.2. The authors used the ex-vivo colon fragments in the experiment. What are the main differences between ex-vivo colon fragments and the vivo ones?
Response 5: Thank you for this interesting question. The main factor that affects the dielecric properties between in-vivo and ex-vivo conditions of the colon fragment is the loss of moisture as the tissue is extracted from the body. To prevent it from happenging we minimized the cold ischemia time by leaving the colon under test as long as possible inside the patient once devitalized. We also minimizee the time between resection and measurement and storee the sample in an airtight container throughout the transport time.
Point 6: Section 3.1. For the data in Table 1 and Figure 4, were them obtained from another place or by measurement?
Response 6: The dielectric properties of the 15 colon samples presented in this paper were measured by the authors using an open-ended coaxial probe.
Please note that we have included an English language and style spell check by a native English-speaking colleague. The revisions are marked up using the “Track Changes” function of MS Word.